# Patient-Centered Communication (PCC) scale: Psychometric analysis and validation of a health survey measure

**Richard P. Moser**[1]*, **Neha Trivedi**[2], **Ashley Murray**[1], **Roxanne E. Jensen**[3], **Gordon Willis**[1], **Kelly D. Blake**[1]

**1** Behavioral Research Program, Division of Cancer Control and Population Sciences, National Cancer Institute, Rockville, MD, United States of America, **2** NORC, at the University of Chicago, Chicago, IL, United States of America, **3** Healthcare Delivery Research Program, Division of Cancer Control and Population Sciences, National Cancer Institute, Rockville, MD, United States of America

☉ These authors contributed equally to this work.
* moserr@mail.nih.gov

## Abstract

### Introduction

Patient-centered communication (PCC) is one important component of patient-centered care and seen as a goal for most clinical encounters. Previous research has shown that higher PCC is related to an increase in healthy behaviors and less morbidity, among other outcomes. Given its importance, the National Cancer Institute (NCI) commissioned a monograph in 2007 to synthesize the existing literature on PCC and determine measurement objectives and strategies for measuring this construct, with a particular focus on cancer survivors. Based on this effort, a seven-item PCC scale was included on the Health Information National Trends Survey (HINTS), a probability-based survey of the US adult population. This study used HINTS data collected in 2018 to evaluate the psychometric properties of the PCC scale for the general US adult population including measures of reliability and validity.

### Results

Through an exploratory factor analysis, the seven-item PCC scale was shown to be unidimensional with good internal consistency (Cronbach's alpha = .92). A confirmatory factor analysis verified the factor structure. Other construct validity metrics included known groups and discriminant validity. Known group comparisons were conducted for several sociodemographic factors and health self-efficacy confirming a priori assumptions. Discriminant validity tests with measures of social support and anxiety/depression showed relatively weak associations.

### Conclusions

The psychometric properties of this scale demonstrate its scientific utility for both surveillance research and other smaller-scale studies. Given its association with many health outcomes, it can also be used to better understand the dynamics in a clinical encounter.

**Data Availability Statement:** All data used for this manuscript are available to the public and can be accessed through this site after accepting a data use agreement: https://hints.cancer.gov/data/

download-data.aspx. This study specifically used the HINTS 5 Cycle 2 data (2018).

**Funding:** There were no external sources of funding for hits project.

## Introduction

Patient-centered care, defined as care that is respectful of and responsive to a patient's preferences and values [1] is seen as one important goal for clinical encounters, and increasing patient-centered communication (PCC) is a way of achieving this goal. Most experts agree that PCC should focus on understanding the patient's perspective and values and supporting the patient in making health decisions that are concordant with these goals [2–4] and thus is seen as an important component of social support [2, 4]. PCC also supports shared decision making [5], which is another important element of healthcare quality.

Research has shown that better PCC is associated with several health outcomes including higher quality of life in cancer survivors [6], higher cancer screening behaviors [7], and other psychological outcomes such as increased health-related self-efficacy, that is, the confidence in one's ability to take care of one's health [8, 9]; lower anxiety, less negative affect, and higher physician trust [10]. Higher PCC in the clinical encounter is also associated with higher self-efficacy for those with multiple chronic health conditions [8] and is critical for patient self-management and a sense of agency in treating one's disease(s). Higher PCC is associated with patient demographics including gender (females), age (older), race-ethnicity (White), and increased healthcare use [9, 11]. PCC is clearly an important factor to assess given its associations with several significant health outcomes and is especially important to understand when treating those with chronic and debilitating diseases to increase self-efficacy given the myriad clinical encounters they will assumedly experience and the need to keep track of their treatment and treatment decisions over the course of the disease.

In 2007 the National Cancer Institute (NCI) commissioned a monograph titled "Patient-Centered Communication in Cancer Care: Promoting Healing and Reducing Suffering" to synthesize the existing literature on PCC, determine measurement objectives and strategies for measuring PCC, and identify gaps in research for cancer survivors in particular [4]. The monograph outlined six critical elements of PCC that are applicable to any clinical encounter: 1) Responding to emotions; 2) Exchanging information; 3) Making decisions; 4) Fostering healing relationships; 5) Enabling patient self-management; 6) Managing uncertainty. It focused on PCC among cancer patients and their significant communication needs given the complex nature of cancer care which oftentimes involves coordination among providers. Related to their cancer treatment, many people also experience negative psychological and physical reactions to cancer treatment [12, 13] and have fatalistic perceptions of a cancer diagnosis [14] and so have other physical, psychological, and emotional considerations that need to be addressed during clinical encounters. However, these issues also apply to other acute and chronic diseases beyond cancer, making PCC an important facet of treatment for all patient-provider encounters.

Given the importance of PCC for many health-related outcomes, the National Cancer Institute's Health Information National Trends Survey (HINTS; https://hints.cancer.gov/) added PCC items to the survey starting in 2008 (HINTS 3), reflecting the core elements of PCC as defined by the NCI monograph [4]. These items have been included on every iteration since HINTS 3 (2008), including the most recent iteration in 2020, and these items have been used in numerous studies as predictors, modifiers, and outcomes. This study will assess the psychometric properties of the PCC scale used in HINTS, and though the psychometric properties of this measure along with other PCC scales have been assessed previously for cancer survivors [15], this is the first study of which we are aware that assesses this PCC scale in a nationally-representative U.S. general adult population. We hypothesized that the psychometric analysis would verify that the PCC scale is unidimensional, with strong internal consistency (reliability), construct validity, with significant positive associations with self-reported health self-efficacy, female gender, education, and age. To demonstrate discriminant validity, given previous

research regarding social support being a component of PCC and PCC being related to anxiety, we hypothesized that PCC scores would be positively, but weakly associated with social support and negatively, but weakly associated with anxiety and depression.

## Materials and methods

### Data source

The test of the psychometric properties of the PCC scale used data from HINTS 5 Cycle 2 gathered from January through May of 2018 with an overall response rate of 32.9% (n = 3,504). Though there have been subsequent iterations of HINTS that could be used, this dataset was one of the most recent when this paper was conceptualized and all HINTS samples are representative of the same population, that is, non-institutionalized US adults eighteen and older. We were also able to take advantage of other existing items on this iteration that could be used to test construct and discriminant validity. HINTS is a probability-based complex sample survey that assesses the American public's use of health and cancer-related information and related knowledge, perceptions, and behaviors in the context of a changing communication environment. Information about the stratified sampling design, creation of the weights, and response rate calculation details are available publicly from the following website: http://hints.cancer.gov/.

HINTS 5 Cycle 2 used an address-based frame. Households were randomly chosen and sent a mail survey wherein the adult (18 or older) with the "Next Birthday" was requested to fill out the survey. A waiver of documentation of informed consent was approved as the research presents no more than minimal risk of harm to subjects and involves no procedures for which written consent is normally required outside of the research context. Further details on the overall design and study operations are published elsewhere. Given that this study used secondary data that were de-identified, it was considered "not human subjects research" according to the NIH and thus did not require an IRB determination.

### Scale items

**Patient-Centered Communication (PCC) scale.** The PCC scale consists of seven items, using concepts from Epstein and Street (4), that probe about the respondents' experience with their provider. The scale states ". . .during the past 12 months. . .", ". . .how often did your doctors, nurses or other health care professionals. . . do the following. . .": 1) "Give you the chance to ask all the health-related questions you had"; 2) "Give the attention you needed to your feelings and emotions"; 3) "Involve you in decisions about your health care as much as you wanted"; 4) "Make sure you understood the things you needed to do to take care of your health"; 5) "Explain things in a way you could understand"; 6) "Spend enough time with you"; 7) "Help you deal with feelings of uncertainty about your health or health care." The PCC scale uses a Likert-like four-point scale: Always (1), Usually (2), Sometimes (3) and Never (4). Scale scores are created by reverse-scoring all items, summing all scores and taking the average and doing a linear transformation to change the range from 0 to 100 with higher scores meaning better communication with one's provider.

The HINTS survey included skip logic, wherein only respondents who saw their provider in the past 12 months completed the PCC scale (n = 2940). A minimum of four valid PCC item responses were needed to generate a PCC scale score.

### Sociodemographic variables

On HINTS the following sociodemographic self-report data were collected and coded into the following categories: Gender (Female/Male); Age (18–35, 36–64; 65+); Race/ethnicity: (Non-

Hispanic White vs. Non-White [includes Latino, Non-Hispanic Black/Asian/Other]). These were used to test for construct validity of the PCC scale.

### Health self-efficacy

Health self-efficacy, an item tested and developed for HINTS, was measured by asking respondents, "Overall, how confident are you about your ability to take good care of your health?" and was dichotomized to Not at all Confident\A Little Confident\Somewhat Confident vs. Very\Completely Confident. This item was used to test for construct validity.

### PROMIS instrumental support measure (4a)

HINTS 5 Cycle 2 included the 4-item PROMIS Instrumental Support Measure and was used to assess the discriminant validity of the PCC scale given its association. This Measure assesses social support and specific functional aspects of these relationships, with an emphasis on health-related support. Items include: 1) "Do you have someone to prepare your meals if you are unable to do it yourself?"; 2) Do you have someone to take you to the doctor if you need it?; 3) Do you have someone to help with your daily chores if you are sick?; 4) Do you have someone to run errands if you need it?, with response options of Never, Rarely, Sometimes, Often, Always (1–5). Raw scores were converted to a T score metric, calibrated to reflect a general population mean of 50 and 10 points reflecting a standard deviation in the general population. Previous research showed the Cronbach's alpha for this scale is α = 0.96 [16]. See for more information: http://www.healthmeasures.net/images/PROMIS/manuals/PROMIS_Instrumental_Support_Scoring_Manual.pdf).

### Patient-Health Questionnaire 4 (PHQ4)

The PHQ4 is found on almost all HINTS iterations and was used to test for discriminant validity of the PCC scale. The PHQ4 was developed by Kroenke et al. [17] as a screening tool to measure anxiety and depression and is included in HINTS. It consists of four items that ask respondents, "Over the past 2 weeks, how often have you been bothered by any of the following problems?" in regards to 1)"Little interest or pleasure in doing things"; 2) "Feeling down, depressed, or hopeless"; 3) "Feeling nervous, anxious, or on edge; and 4) "Not being able to stop or control worrying". Response options include "Nearly every day" (1), "More than half the days" (2), "Several days" (3), and "Not at all" (4). Scale scores are created by first rescoring the items from 0–3, reverse scoring the items and then summing them, thus total scores range from 0–12 with higher scores indicating more depression and anxiety. The PHQ4 has been shown to have good internal reliability, construct validity, and factorial validity [17].

### Statistical analysis

We first computed weighted descriptive statistics of the respondents. Next, unweighted descriptive statistics for the PCC scale items and scale were calculated, including measures of central tendency. Aspects of data quality were measured through the percent missing by item. The internal consistency (a measure of reliability) of the PCC scale was determined by examining item-total correlations, Cronbach's alpha, and an exploratory factor analysis. Next, construct validity was measured through a confirmatory factor analysis (CFA) of a one-factor structure—with the variance fixed to 1 to free the factor loadings—and verifying known-group differences between variables previously assessed in the literature [15] and included on HINTS 5 Cycle 2 including health self-efficacy, race/ethnicity, gender and age. For the CFA, fit was determined using the following fit indices: Root Mean Square Error of Approximation

(RMSEA, cut off < .08), Comparative Fit Index (CFI, cut off >.90), Tucker Lewis Index (TLI, cut off >.95), and Standardized Root Mean Square Residual (SRMR, cut off < .08). We did not report the chi square statistic fit as research has found that this test is sensitive to large sample sizes. The known-group differences analysis was done with weighted t-tests and means are presented to illustrate any differences. Lastly, discriminant validity was calculated by examining the Pearson correlation between the PCC scale and the PROMIS social support scale and the PHQ4. Discriminant validity is shown by having associations, albeit weakly, between PCC and the other scales [18] Note that p-values are not typically reported for these analyses and interpretation of a significant p value (i.e., $p < .05$) could be misleading given the relatively large sample size used here.

## Results

### Descriptive statistics

See Table 1 for a description of the HINTS 5 Cycle 2 sample used in this analysis. The overall PCC scale mean was 79.75 (range 0–100) with a median of 85.7 and skewness = -.95. Note that 33% of the sample had a PCC score of 100, the highest score possible, indicating a ceiling effect.

### Reliability

Internal consistency was assessed by calculating Cronbach's Alpha for PCC scores and was $\alpha$ = 0.92. Descriptive statistics (mean, % missing) for each item and the item-total score correlations and Cronbach's alpha if item deleted can be seen in Table 2. Results of the exploratory factor analysis showed one factor through the scree plot which was further supported by an eigenvalue greater than 1 (factor 1 = 12.34; no other factors had an eigenvalue greater than 1). Factor loadings (for one factor) ranged from .78 to .83.

**Table 1. Unweighted cell sizes and weighted percentages of the HINTS 5 Cycle 2 sample (n = 2940) of those who visited their provider in the last 12 months and had a valid PCC score.**

| Characteristic | Sample Size | Percentage (95% CI) |
|---|---|---|
| **Gender** | | |
| Male | 1121 | 45.2 (43.8, 46.6) |
| Female | 1777 | 54.7 (53.4, 56.1) |
| **Age** | | |
| 18–35 | 344 | 22.8 (20.3, 25.3) |
| 36–50 | 555 | 28.4 (25.8, 30.9) |
| 51–64 | 880 | 27.2 (25.0, 29.3) |
| 65+ | 1095 | 21.6 (20.7, 22.5) |
| **Education** | | |
| Less than high school | 210 | 7.3 (5.5, 9.0) |
| High school graduate | 512 | 21.2 (19.1, 23.3) |
| Some college | 871 | 40.2 (37.9, 42.4) |
| College grad or more | 1309 | 31.3 (30.1, 32.6) |
| **Race/Ethnicity** | | |
| Non-Hispanic White | 1739 | 67.6 (66.1, 69.2) |
| Non-Hispanic Black/AA | 367 | 10.3 (9.3, 11.3) |
| Non-Hispanic Asian | 105 | 5.1 (4.1, 6.2) |
| Hispanic | 343 | 13.3 (12.2, 14.4) |
| Non-Hispanic Other/Multiple Races | 107 | 3.6 (2.9, 4.3) |

**Table 2. PCC scale items, means, % missing, standardized variables item/total correlations and alpha if item deleted.**

| PCC Scale Item* | Mean (range = 1–4) | % Missing | Correlation with Item Total | Cronbach's Alpha if Deleted |
|---|---|---|---|---|
| Questions | 3.54 | 1.62 | 0.74 | 0.920 |
| Attention | 3.25 | 2.03 | 0.78 | 0.917 |
| Decisions | 3.41 | 2.03 | 0.79 | 0.916 |
| Understood | 3.54 | 1.93 | 0.80 | 0.915 |
| Explain | 3.59 | 2.03 | 0.76 | 0.919 |
| Time | 3.26 | 2.33 | 0.78 | 0.916 |
| Feelings | 3.17 | 2.57 | 0.76 | 0.919 |

* Full text of item

Question: "Give you the chance to ask all the health-related questions you had"

Attention: "Give the attention you needed to your feelings and emotions"

Decisions: "Involve you in decisions about your health care as much as you wanted"

Understood: "Make sure you understood the things you needed to do to take care of your health"

Explain: "Explain things in a way you could understand"

Time: "Spend enough time with you"

Feelings: "Help you deal with feelings of uncertainty about your health or health care."

## Validity

**Construct validity.** *Confirmatory factor analysis.* Results of the EFA supported the unidimensionality of the PCC scale, so a one-factor confirmatory factor analysis was conducted to test the fit of this model. Factor loadings and model fit statistics from this analysis can be found in Table 3 (with cut-offs indicating good fit). Overall, the CFA indicated good fit for a 7-item unidimensional scale.

## Known group differences

Previous research with data other than HINTS has shown group differences in PCC scores regarding health self-efficacy, gender, race/ethnicity, and age with PCC being positively associated with higher values of self-efficacy, being female, White and being older. This analysis assessed for these hypothesized group differences using HINTS. T-tests showed that the following variables exhibited significant between group differences in the hypothesized direction:

**Table 3. Standardized factor loadings and fit statistics for confirmatory factor analysis of the PCC scale.**

| Item | Factor Loading |
|---|---|
| Give you the chance to ask all the health-related questions you had | .858 |
| Give the attention you needed to your feelings and emotions | .877 |
| Involve you in decisions about your health care as much as you wanted | .884 |
| Make sure you understood the things you needed to do to take care of your health | .919 |
| Explain things in a way you could understand | .900 |
| Spend enough time with you | .877 |
| Help you deal with feelings of uncertainty about your health or health care | .867 |

Fit indices (cut-off for good fit)

RMSEA (Root Mean Square Error Of Approximation): 0.118 (< .08)

CFI (Comparative Fit Index): 0.991 (>.90)

TLI (Tucker Lewis Index): 0.986 (>.95)

SRMR (Standardized Root Mean Square Residual): 0.022 (< .08)

**Table 4. Known-group comparisons, weighted means, standard errors, mean group differences and probabilities for PCC scale scores (Significant differences, alpha = .05, bolded).**

| Known-Group Comparison | Hypothesized Higher PCC Scores | Group 1 Mean (SE) | Group 2 Mean (SE) | Mean Group Difference (SE) | P Value |
|---|---|---|---|---|---|
| Health Self-Efficacy (High vs. Low Confidence) | High Confidence | 82.19 (.63) | 75.80 (1.32) | 6.40 (1.32) | **< .0001** |
| Age 18–35 vs. 65+ | 65+ | 78.41 (1.69) | 81.76 (.98) | -3.35 (1.80) | .069 |
| Female vs. Male | Female | 80.90 (.83) | 79.52 (.97) | 1.37 (1.29) | .293 |
| White vs. Non-White | White | 81.48 (.78) | 77.50 (1.35) | 3.98 (1.64) | **.019** |

health self-efficacy (high confidence > low confidence) and race/ethnicity (White > Non-White); groups defined by age and gender, though trending in the expected direction, were not significantly different. Table 4 presents the weighted means, standard errors, mean differences and p values for each group comparison.

**Discriminant validity.** As mentioned previously, discriminant validity is shown by having associations, albeit weakly, between PCC and other theoretically related items [18]. For this analysis, we were able to utilize two existing scales found on HINTS 5 Cycle 2, Instrumental Support (PROMIS) and the PHQ4. Results of the Pearson correlation between PCC and Instrumental Support showed the two scales were weakly correlated (Pearson correlation = .19). The Pearson correlation between PCC and the PHQ4 showed a negative association that was also relatively weak (-.14). The directions of the associations were in the hypothesized direction.

## Discussion

Based on a conceptual model and utilizing a nationally representative sample of US adults, the psychometric properties of the Patient-Centered Communication scale were assessed and found to show good reliability and validity. Several of the expected group differences were found, with education and age as notable exceptions. Regarding measures of data quality, across items there was a small amount of missing data seemingly showing that people understood and were comfortable answering each question though there was a tendency for respondents to answer "Always" for each PCC item. This outcome may signify satisficing, that is, putting in minimal effort to answer the questions, but a better explanation may be that the PCC scale is showing some ceiling effects and difficulty with discerning differences at the higher end of the scale, which is a limitation of this scale. Ceiling effects are oftentimes seen in patient-provider communication scales [19] and other measures of healthcare satisfaction like the Consumer Assessment of Healthcare Providers and Systems (CAHPS) [20]. An alternative interpretation is that most adults are at least somewhat satisfied with their clinical encounters, and it would be useful to replicate these results with other data and especially with data from providers regarding their usual practice. We are not aware of other data sources with information on providers regarding this practice. Another mitigating factor that lessens concerns is that the PCC showed a minimal floor effect (i.e., about .2% of respondents scored the lowest value on the PCC scale), so this measure is particularly sensitive to communication deficits.

Given the rigorous study of patient-provider communication and strong conceptual model on which the PCC scale was designed, it was not surprising to confirm the association of the individual components and the overall unidimensional nature of the measure, which is consistent with other similar analyses [15]. Fit indices were good for almost all metrics. Expected group differences in self-efficacy and race/ethnicity make conceptual sense and replicated previous research. A lack of group differences by gender and age was unexpected, although the

trend for females and older respondents to have higher PCC was observed (though not statistically significant). Perhaps these variables interact with other variables in explaining PCC differences and future research can test this out. Discriminant validity indices, showing weak associations suggest that though there is overlap in variance explained between PCC and the other scales, the PCC scale also appears to explain unique variance which supports its use as an independent variable.

This study had several limitations. Responses were self-reported and may be subject to a variety of cognitive biases. The PCC scale also provides an overall perception of clinical encounters during the past 12 months as opposed to focusing on any specific provider and so may be interpreted as a gestalt view of a typical encounter. Given the cross-sectional nature of the survey, we couldn't assess other aspects of reliability such as test-retest. Lastly, there were not enough responses for Spanish speakers so it's not clear if the results would generalize to other languages.

## Clinical implications

Given the increased focus on the "activated" patient and the importance of shared decision making between patient and provider, the PCC scale has practical applicability for both public health and clinical researchers and patient advocates and providers. For researchers, understanding prevalence of PCC at a population or sub-population level (e.g., state or county; specific racial/ethnic groups) and assessing for any correlates or causes (e.g., social determinants of health) will provide ideas for clinicians or patient advocates to intervene in certain geographic areas or with specific groups. A PCC assessment could be embedded in health systems and assessed regularly similar to how smoking status is now asked. Once identified, educating patients about the importance of asking questions and making sure emotional needs are addressed can be stressed. Likewise, it is critical that providers receive education in good communication skills as research has shown positive patient outcomes associated with better provider communication [21] and so professional organizations have advocated for this type of training in medical education.

## Conclusions

In summary, the PCC scale showed strong internal consistency and good construct validity and discriminant validity. The scale is unidimensional though it did show some ceiling effects similar to other communication measures. It did not show floor effects and seems particularly useful for assessing deficits in patient-provider communication. PCC continues to be included in HINTS iterations and future research could test for trends in this outcome over time and examine any differential change by important sociodemographic variables such as gender, race/ethnicity, and age. Other validation could be done with cut-points or percentiles to further elucidate the utility of the scale. The PCC scale could be included in other population-level health surveys like HINTS to compare across surveys or to test with targeted populations. Lastly, this is also a relatively short scale that could be incorporated into the clinical setting to assess this important construct.

## Acknowledgments

**Disclaimers:** The article was prepared as part of the authors' official duties as employees or fellows of the US Federal Government. The findings and conclusions in this report are those of the authors and do not necessarily represent the official position of the National Cancer Institute.

## Author Contributions

**Conceptualization:** Richard P. Moser, Neha Trivedi, Ashley Murray, Roxanne E. Jensen, Gordon Willis, Kelly D. Blake.

**Formal analysis:** Richard P. Moser, Neha Trivedi, Ashley Murray, Roxanne E. Jensen.

**Methodology:** Richard P. Moser, Neha Trivedi, Ashley Murray, Roxanne E. Jensen, Gordon Willis, Kelly D. Blake.

**Writing – original draft:** Richard P. Moser, Neha Trivedi, Ashley Murray, Roxanne E. Jensen, Gordon Willis, Kelly D. Blake.

**Writing – review & editing:** Richard P. Moser, Neha Trivedi, Ashley Murray, Roxanne E. Jensen, Gordon Willis, Kelly D. Blake.

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
