## [Decision Letter · Decision Letter 0]

14 Sep 2022

PONE-D-22-06849Patient-Centered Communication (PCC) Scale:  Psychometric Analysis and Validation of a Health Survey MeasurePLOS ONE

Dear Dr. Moser,

Thank you for submitting your manuscript to PLOS ONE. After careful consideration, we feel that it has merit but does not fully meet PLOS ONE’s publication criteria as it currently stands. Therefore, we invite you to submit a revised version of the manuscript that addresses the points raised during the review process.

Your manuscript has been assessed by two peer-reviewers and their reports are appended below.  The reviewers comment that the manuscript could be strengthened by including a discussion on how the ceiling effect reported in this study compares to the ceiling affects reported in other similar studies. In addition, the reviewers request that more information is provided on ow the CFA fit was evaluated. Could you please carefully revise the manuscript to address all comments raised?

We look forward to receiving your revised manuscript.

Kind regards,

Maria Elisabeth Johanna Zalm, Ph.D

Editorial Office

PLOS ONE

Journal Requirements:

2. Please include your tables as part of your main manuscript and remove the individual files. Please note that supplementary tables (should remain/ be uploaded) as separate "supporting information" files".

Reviewers' comments:

Reviewer's Responses to Questions

**Comments to the Author**

1. Is the manuscript technically sound, and do the data support the conclusions?

Reviewer #1: Yes

Reviewer #2: Yes

2. Has the statistical analysis been performed appropriately and rigorously? 

Reviewer #1: Yes

Reviewer #2: Yes

3. Have the authors made all data underlying the findings in their manuscript fully available?

Reviewer #1: Yes

Reviewer #2: Yes

4. Is the manuscript presented in an intelligible fashion and written in standard English?

Reviewer #1: Yes

Reviewer #2: Yes

5. Review Comments to the Author

Reviewer #1: This is a really nicely done study that is written clearly. I have only a minor comment. As the measure has such a pronounced ceiling effect (33% of respondents having a "perfect" score, I was hoping the authors could comment on how this compares to ceiling effects of other patient-reported measures of provider communication, such as the CAT (Makoul) or the CAHPS communication items (AHRQ). Have the authors considered what cutpoints might be useful (e.g., perfect scores against all else; three tiers perfect, good, inadequate) to deal with these ceiling effects?

Reviewer #2: This high quality manuscript describes the psychometric validation of a 7-item patient-centered communication scale in a large nationally representative sample of U.S. adults. Due to the paucity of well-validated communication scales, this work is a valuable contribution.

No major concerns. This study appropriately evaluated the psychometric properties of the instrument as could be done with the available data.

Minor concerns: Information about how the CFA fit was to be evaluated (“For the CFA, fit was

determined using the following fit indices: Root Mean Square Error of Approximation (RMSEA,

cut off <.08), Comparative Fit Index (CFI, cut off >.90), Tucker Lewis Index (TLI, cut off >.95),

and Standardized Root Mean Square Residual (SRMR, cut off <.08). We did not report the chi

square statistic fit as research has found that this test is sensitive to large sample sizes”) should be provided in the Methods section instead of Results. Similarly, in the section on discriminant validity, for the point about not reporting p-values for correlation coefficients.

6. PLOS authors have the option to publish the peer review history of their article (what does this mean?). If published, this will include your full peer review and any attached files.

Reviewer #1: No

Reviewer #2: No

---

## [Author Response · Author response to Decision Letter 0]

4 Nov 2022

Note this information was also included in a separate document uploaded previously. 

Dear Editor: Thanks for the opportunity to respond to the reviewers’ comments. Below you’ll see the comments and our response to each point in red.

Review Comments to the Author

Reviewer #1: This is a really nicely done study that is written clearly. I have only a minor comment. As the measure has such a pronounced ceiling effect (33% of respondents having a "perfect" score, I was hoping the authors could comment on how this compares to ceiling effects of other patient-reported measures of provider communication, such as the CAT (Makoul) or the CAHPS communication items (AHRQ). Have the authors considered what cutpoints might be useful (e.g., perfect scores against all else; three tiers perfect, good, inadequate) to deal with these ceiling effects?

Response: Thank you for this comment. We agree that the PCC showed higher than ideal ceiling effects. However, due to the communication-focused construct, and number of items, this was not unexpected. For example, the PCC ceiling effect is comparable to CAHPS (we’re not familiar with the CAT measure). We have added a reference and additional information about this on pages 15 and 16. Fortunately, the PCC also reported a minimal floor effect (0.2%). Therefore, we can feel comfortable to conclude this validation is sensitive to communication deficits. This has been noted both in the Discussion and Conclusion sections. 

We also agree that PCC cut-points and percentiles would be incredibly useful to examine communication-focused research questions. However, our psychometric analyses were designed to evaluate the scale’s item-level properties, and an overall evaluation of the total scale score. We strongly encourage investigators to consider common cut-points and percentiles, based on their proposed research questions and target population and added this information to the conclusion. 

Reviewer #2: This high quality manuscript describes the psychometric validation of a 7-item patient-centered communication scale in a large nationally representative sample of U.S. adults. Due to the paucity of well-validated communication scales, this work is a valuable contribution.

Response: We’re very happy to hear this is considered a valuable contribution. 

No major concerns. This study appropriately evaluated the psychometric properties of the instrument as could be done with the available data.

Response: We appreciate this positive comment.

Minor concerns: Information about how the CFA fit was to be evaluated (“For the CFA, fit was

determined using the following fit indices: Root Mean Square Error of Approximation (RMSEA,

cut off <.08), Comparative Fit Index (CFI, cut off >.90), Tucker Lewis Index (TLI, cut off >.95),

and Standardized Root Mean Square Residual (SRMR, cut off <.08). We did not report the chi

square statistic fit as research has found that this test is sensitive to large sample sizes”) should be provided in the Methods section instead of Results. Similarly, in the section on discriminant validity, for the point about not reporting p-values for correlation coefficients.

Response: The information noted here was moved from the Results to the Statistical Analysis subsection on pages 9-10 in the Methods section.

---

## [Decision Letter · Decision Letter 1]

14 Dec 2022

Patient-Centered Communication (PCC) Scale:  Psychometric Analysis and Validation of a Health Survey Measure

PONE-D-22-06849R1

Dear Dr. Moser,

We’re pleased to inform you that your manuscript has been judged scientifically suitable for publication and will be formally accepted for publication once it meets all outstanding technical requirements.

Kind regards,

Nabi Nazari, PhD

Academic Editor

PLOS ONE

Additional Editor Comments (optional):

Reviewers' comments:

Reviewer's Responses to Questions

**Comments to the Author**

1. If the authors have adequately addressed your comments raised in a previous round of review and you feel that this manuscript is now acceptable for publication, you may indicate that here to bypass the “Comments to the Author” section, enter your conflict of interest statement in the “Confidential to Editor” section, and submit your "Accept" recommendation.

Reviewer #1: All comments have been addressed

Reviewer #2: All comments have been addressed

2. Is the manuscript technically sound, and do the data support the conclusions?

Reviewer #1: Yes

Reviewer #2: Yes

3. Has the statistical analysis been performed appropriately and rigorously? 

Reviewer #1: Yes

Reviewer #2: Yes

4. Have the authors made all data underlying the findings in their manuscript fully available?

Reviewer #1: Yes

Reviewer #2: Yes

5. Is the manuscript presented in an intelligible fashion and written in standard English?

Reviewer #1: Yes

Reviewer #2: Yes

6. Review Comments to the Author

Reviewer #1: The comments have been addressed adequately. My sincere thanks to the authors for their responsiveness.

Reviewer #2: (No Response)

7. PLOS authors have the option to publish the peer review history of their article (what does this mean?). If published, this will include your full peer review and any attached files.

Reviewer #1: No

Reviewer #2: No

---

## [Editor Report · Acceptance letter]

20 Dec 2022

PONE-D-22-06849R1 

Patient-Centered Communication (PCC) Scale:  Psychometric Analysis and Validation of a Health Survey Measure 

Dear Dr. Moser:

I'm pleased to inform you that your manuscript has been deemed suitable for publication in PLOS ONE. Congratulations! Your manuscript is now with our production department. 

Kind regards, 

on behalf of

Dr. Nabi Nazari 

Academic Editor

PLOS ONE